# Functional Toll-Like Receptors (TLRs) Are Expressed by a Majority of Primary Human Acute Myeloid Leukemia Cells and Inducibility of the TLR Signaling Pathway Is Associated with a More Favorable Phenotype

**DOI:** 10.3390/cancers11070973

**Published:** 2019-07-11

**Authors:** Annette K. Brenner, Øystein Bruserud

**Affiliations:** 1Department of Medicine, Haukeland University Hospital, 5021 Bergen, Norway; 2Section for Hematology, Department of Clinical Science, University of Bergen, 5020 Bergen, Norway

**Keywords:** acute myeloid leukemia, toll-like receptors, cytokine, epigenetic modification, survival

## Abstract

Acute myeloid leukemia (AML) is a highly heterogeneous disease with regard to biological characteristics and receptor expression. Toll-like receptors (TLRs) are upstream to the transcription factor NFκB and part of the innate immune system. They are differentially expressed on AML blasts, and during normal hematopoiesis they initiate myeloid differentiation. In this study, we investigated the response upon TLR stimulation in an AML cohort (*n* = 83) by measuring the increase of NFκB-mediated cytokine secretion. We observed that TLR4 is readily induced in most patients, while TLR1/2 response was more restricted. General response to TLR stimulation correlated with presence of *nucleophosmin* gene mutations, increased mRNA expression of proteins, which are part of the TLR signaling pathway and reduced expression of transcription-related proteins. Furthermore, signaling via TLR1/2 appeared to be linked with prolonged patient survival. In conclusion, response upon TLR stimulation, and especially TLR1/2 induction, seems to be part of a more favorable phenotype, which also is characterized by higher basal cytokine secretion and a more mature blast population.

## 1. Introduction

Acute myeloid leukemia (AML) is an aggressive malignancy which is characterized by bone marrow infiltration of immature leukemic cells. Even though AML is the most common acute leukemia, it still is a relatively rare disease with a late onset in life [1,2]. Most cases occur without apparent cause, but AML can also be secondary to for instance myelodysplastic syndromes (MDS) or chemotherapy [3,4]. The disease is highly heterogeneous with regard to cell morphology, cytogenetics and gene mutations, of which especially the latter two influence prognosis and, ultimately, overall survival [5].

Toll-like receptors (TLRs) are part of the innate immune system and detect pathogen- and danger associated molecular patterns derived from bacteria, viruses, fungi or parasites [6]. In humans, ten TLRs have been identified so far [7]. The TLRs differ in their localization as TLR1/2/4–6/10 are situated at the plasma membrane, whereas TLR3/7–9 and partially TLR4 localize to membranes of cytosolic compartments [8]. Upon ligand binding, TLRs homo- or heterodimerize which leads to recruitment of adaptor proteins—with MyD88 being the most abundant as it is shared by all TLRs but TLR3 [9]—and the subsequent initiation of signaling cascades [8]. The latter have a common target in the transcription factor NFκB which upon activation leads to the transcription of pro-inflammatory cytokines [7,10].

Previous studies suggest that TLRs are expressed by various hematological malignant cells, including AML cells, and TLR agonists may therefore have direct effects on the leukemic cells [11]. First, primary AML cells seem to have increased levels of TLR2 and TLR4 [12], which then are associated both with chemoresistance and monocytic differentiation [13]. Second, several TLRs are also expressed in AML cell lines and specific receptor agonists seem to employ growth-inhibitory and pro-apoptotic effects in some of the cell lines [13,14]. Third, the dual TLR7/8 agonist R848 inhibited the growth of human AML cells in immunodeficient mice through a direct effect on the AML cells [15]. TLRs are also expressed by AML stem cells, and TLR1/2 ligation stimulates stem cell growth [16]. Thus, whether TLR ligation causes growth inhibition or stimulation seems to depend on the experimental model, and the results are also conflicting whether TLR ligation can induce leukemic cell differentiation [15,17]. TLRs are furthermore regarded as possible drug targets in AML due to indirect antileukemic effects caused by the activation/stimulation of normal immunocompetent cells, e.g., for eradication of residual leukemic cells after induction therapy [18]. Furthermore, the release of pro-inflammatory cytokines as a result of TLR stimulation can enhance the immunogenicity of AML blasts, rendering them more susceptible towards treatment [19].

As outlined above, previous studies of TLRs in AML have mainly focused on cell proliferation and survival, and several of the studies were mainly based on the use of cell lines. In the present study, we therefore focused on detection of functional receptors (i.e., protein expression), mRNA expression of TLRs and proteins of the downstream signaling pathway, communication with neighboring non-leukemic cells (i.e., cytokine release) and associations between TLR responsiveness and genomic profiles. None of these aspects have been addressed in previous studies on a large and unselected patient cohort. We found TLR response to be correlated with *nucleophosmin* (*NPM1*) mutations, increased mRNA expression of proteins of the TLR signaling pathway, and low expression of transcription-related proteins. Furthermore, especially signaling via the TLR1/2-NFκB axis appeared to be associated with an improved outcome.

## 2. Results

### 2.1. AML Patients Included in the Study

We investigated primary human AML cells derived from 83 consecutive patients admitted to the Section for Hematology, Haukeland University Hospital, which is responsible for diagnosis and clinical handling of all AML patients within a defined geographical area. The cell samples were derived from patients with relatively high levels of circulating leukemic cells. The cells were stored in the public biobank of the Section for Hematology. Diagnostic information about each sample was available from the biobank (i.e., karyotype and mutational characteristics, FAB classification based on histochemistry, flow cytometric analyses of differentiation markers, results of treatment) together with the enrichment of leukemic cells in the samples. All experiments were based on cryopreserved viable cells; all cells were collected, prepared, cryopreserved and thawed according to highly standardized protocols. All samples were prepared by density gradient separation (see Section 4.1.). Thus, our patients should be regarded as a population-based and unselected cohort of AML patients with relatively high levels of circulating blasts to allow for preparation of highly enriched AML cell populations based on gradient separation alone. Most contaminating cells (usually <5%) were small lymphocytes. All cell samples included in the present experiments were the total primary AML cell population collected after gradient separation alone without further enrichment or stimulation.

### 2.2. Lipopolysaccharide Readily Induces Cytokine Expression

As expected, inducing TLR receptors with agonists resulted in increased secretion of soluble mediators. CCL2 and IL-6 were the cytokines, whose secretion levels highly increased after addition of all four agonists (Figure 1 and Appendix A). Flagellin (TLR5 agonist) and lipopolysaccharide (LPS; TLR4 agonist) showed the greatest effect on mediator secretion, where the latter resulted in exceptional high increase in the chemokines CCL2/3 and CXCL1, the interleukins IL-1β and IL-6, and additionally tumor necrosis factor α (TNFα) and granulocyte colony-stimulating factor (G-CSF; Figure 1). Only the protease inhibitors cystatin C and serpin E1 were little affected by LPS attribution, which was expected as TLR signaling foremost leads to the expression of pro-inflammatory cytokines. In the Figure, we show the increase in mediator release as the median fraction between samples with and without TLR agonists. We used the median values (i.e., the median effect on AML cells) for each single mediator in order to visualize the effects of all agonists on all soluble mediators in a single figure.

### 2.3. Response towards Agonists Is Linked with Mutations in the Nucleophosmin Gene

Approximately 70% of the 83 patient cell cultures showed a significant increase in mediator secretion after stimulation with TLR agonists. TLRs ligation can initiate intracellular signaling through their main pathway directly down to NFκB [20], but other pathways that either are additionally activated or show crosstalk with TLR-NFκB may also influence mediator release [21]. However, NFκB is documented as an important regulator of the release of several soluble mediators. For these reasons, we decided prior to beginning the study to use cytokine release as the readout for responsiveness to TLR agonists, and to quantify this responsiveness not only by the effects on a single mediator. First, based on the 10 unselected test patients (see Section 4.2.) we defined a significantly altered level for a single mediator as: (i) at least doubling in concentrations for control values > 100 pg/mL, or (ii) an increase of at least 100 pg/mL for values below that threshold. Second, we defined a positive response to a single agonist as significantly increased levels of at least 1/3 (i.e., seven or more) of the mediators and a partial response as significantly altered levels of at least two mediators. Finally, a positive response towards general TLR stimulation was defined as either increase in at least 7/19 mediators for at least two agonists, or an increase of at least 7/19 mediators in one agonist and partial response (2–6 mediators) in at least two additional agonists. Partial response was then defined as partial response to stimulation of at least two agonists. Using these definitions, 57 of the patients were considered to be responders, 10 to be partial responders and 16 to be non-responders to TLR stimulation.

TLR responsiveness appeared to be independent of most AML prognostic factors or patient characteristics. Patient age, gender, disease etiology and *FMS-related tyrosine kinase*-internal tandem repeats (Flt3-ITDs) did not significantly correlate with response upon TLR stimulation. However, we saw that cells with *NPM1* mutations were associated with higher likelihood to be induced by TLR agonists, except for R848 (TLR7/8 agonist). Only 1/23 (4%) patients with an *NPM1* insertion did not respond or showed only partial response to the group of agonists as compared to 20/44 (45%) patients lacking these mutations (χ^2^ likelihood ratio, *p* < 0.001; the first diagnosis data were used for patients with two samples). The same pattern was observed for the single agonists: Pam3CSK4 (TLR1/2 agonist) showed 41% unresponsive patients with *NPM1*-mutations as compared to 81% of patients without (*p* = 0.006). The corresponding values for LPS (TLR4 agonist) and flagellin (TLR5 agonist) were *p* = 0.011 (4% vs. 32%) and *p* = 0.033 (14% vs. 45%). We also saw a correlation between general TLR responsiveness and CD34^-^ cells (62% vs. 24% for CD34^+^ cells; *p* = 0.012), which most likely is caused by the close association between *NPM1* insertions and CD34^-^ cells. Furthermore, it appeared as if cytogenetics correlated with R848 (TLR7/8 agonist) response: 23/45 patients (45%) with normal or favorable (i.e., core-binding factor AML) karyotype responded with increased cytokine secretion upon R848 stimulation as opposed to 7/28 patients (25%) with an intermediate or adverse karyotype (Fisher’s exact test, *p* = 0.010). Lastly, borderline correlation for monocytic differentiation according to the FAB classification and TLR response was also seen for flagellin (TLR5 agonist, *p* = 0.032) and Pam3CSK4 (TLR1/2 agonist, *p* = 0.056). However, we want to emphasize that patient subgroups are relatively small; these results should therefore be interpreted with great care because the number of patients will only allow us to detect the strongest associations.

### 2.4. Response towards the TLR1/2 Agonist Is Linked with Prolonged Patient Survival

Our samples were collected from consecutive AML patients, and our cohort therefore has a high median age and includes several elderly and unfit patients. For this reason, our cohort should be regarded as representative for the AML population, but at the same time only a subset of our patients received intensive remission-inducing and potentially curative treatment. A total of 41 patients received intensive treatment; two of these patients were lost from follow-up and one died from toxicity 26 days after start of induction therapy. We therefore investigated the associations between TLR agonist responsiveness and overall survival for the remaining 38 patients. By that, we have a study population where the patients either died from chemoresistant AML (primary refractory disease or relapse) or survived without relapse after a median follow-up time of 92 months. Death due to toxicity/comorbidity thus did not influence our comparison of in vitro AML cell biology and clinical AML cell chemosensitivity.

Pam3CSK4, the TLR1/2 heterodimer agonist, was to a lesser degree—both quantitatively and qualitatively—than the other three agonists able to induce mediator expression. Only 33% of patients (26/78; samples for five patients were missing from Pam3CSK4 analysis) were responders to TLR1/2 induction, whereas 40% (31/78 patients) were unresponsive to Pam3CSK4. Importantly, patient survival was associated with TLR responsiveness (*p* = 0.037; Appendix A) as median survival for (partial) responders was 2.3 years (95% CI: 1.19–3.31 years) as opposed to 0.58 years (95% CI: 0.16–1.01 years) for non-responders. However, one has to be careful with the interpretation of these data, as only 7/38 patients were in the non-responder group. Of the single agonists, only response upon TLR1/2 stimulation was associated with prolonged patient survival (*p* = 0.045; Figure 2). The median survival time for the 23 (partial) responders was 2.4 years as compared to only 0.67 years (95% CI: 0.26–1.08 years) for the 13 non-responders. Interestingly, two of the three patients—one of them received potential curative treatment and was included in the survival statistics—with samples from both first diagnosis and relapse changed from being responders at first diagnosis to being non-responders at relapse. Because AML relapse is associated with a more aggressive disease, being non-responsive towards TLR induction might be part of a less favorable phenotype.

Because we observed that mutations in *NPM1* and partially cytogenetics correlated with TLR response, we additionally performed Cox regression for uni- and multivariate analysis, including the terms TLR response, patient age (younger or older than 60 years), disease etiology (relapse/secondary vs. de novo AML), *NPM1* insertions, *Flt3*-ITD, and normal/favorable vs. intermediate/adverse karyotype. Both older patient age (crude hazard ration, HR = 1.82; *p* = 0.006) and borderline lack of TLR1/2 responsiveness (HR = 1.52; *p* = 0.057) arose as negative prognostic factors in the univariate analyses. However, only lack of TLR1/2 induction (HR = 1.86; *p* = 0.029) emerged as independent negative prognostic factor in the multivariate Cox regression (Table 1). Because mutation status is missing for four patients (one TLR1/2 responsive patient; one long-term survivor), the latter analyses only contained 32 out of the 36 patients with survival data.

### 2.5. Response upon TLR Activation Is Correlated with Increased Expression of Genes in the TLR Signaling Pathway and High Constitutive Cytokine Release

We searched AmiGo 2 [22] for gene ontology (GO) terms related to “TLR signaling” and identified 129 distinct human proteins. After median-normalization, we clustered the mRNA expression of these 129 proteins obtained from microarray analysis (see Section 4.4) in 46 unselected patients (the patients are indicated in Appendix A). The main subgroup that determined the patient distribution in the cluster included 38 differentially expressed proteins (Figure 3). These 38 proteins contained six TLRs (among them the targeted TLR1, TLR4, TLR5, TLR7 and TLR8), important co-receptors, such as CD14, and adaptors, such as MyD88 (Table 2). But we also found proteins upstream to TLRs, like TLR chaperones and the intrinsic TLR agonists S100A8 and A9. Furthermore, we identified three positive and 13 negative pathway regulators. Among the latter are five inhibitors of NFκB: ARRB2 and HAVCR2 attenuate NFκB mediated transcription, whereas NFKBIA, PIK3AP1 and TANK inhibit NFκB activation. Thus, the differentially expressed genes belonged to all steps in the TLR signaling pathway.

The 46 patients could be categorized into two groups: 13 patients with generally increased mRNA levels, and 33 patients with low mRNA TLR pathway expression. In the high expression group, responders to general TLR activation (12/13 vs. 19/33 patients; Fisher’s exact test: *p* = 0.035) and especially TLR1/2 activation (9/12 vs. 8/30 patients; Fisher’s exact test: *p* = 0.006) were overrepresented. Therefore, response to TLR targeting is at least partly reflected at the mRNA level. Interestingly, 12/13 differentially expressed negative pathway regulators were also found to be upregulated in the TLR responders, indicating that these cells have the ability to counteract pro-longed TLR pathway activation.

We further analyzed whether high basal constitutive release correlated with the TLR response in Figure 3; these analyses were based on the mediators with large concentration ranges and showing high increase in response to TLR activation, namely all chemokines, IL-1β, IL-6 and TNFα. Patients with release above 1 ng/mL in at least two mediators were defined to have high basal levels, whereas patients with undetectable values (usually corresponding to less than 5 pg/mL) in at least five mediators were defined to have low constitutive release. We first compared the high mRNA level patients (*n* = 13) with the patients in the other main subset (*n* = 33). Eight out of 13 patients in the high expression group then presented with high basal cytokine levels without TLR stimulation as compared to only 5/33 patients in the low expression group (Fisher’s exact test: *p* = 0.003). Thus, the mRNA levels of TLR signaling components also seem to reflect the cells’ ability to constitutively produce cytokines. Across the groups in Figure 3, high basal cytokine secretion was seen in 12/31 patients with high TLR responsiveness, whereas only one TLR responder did not show high cytokine release (Fisher’s exact test, *p* = 0.035). The corresponding fractions for TLR1/2 response were 9/17 vs. 4/25 patients (Fisher’s exact test, *p* = 0.018). Thus, a strong responsiveness to TLR agonists was generally associated with high constitutive release of soluble mediators. Further, cells with monocytic cell differentiation (FAB M4/M5) were overrepresented among the high TLR mRNA expression group (10/12 vs. 12/31 patients; Fisher’s exact test: *p* = 0.016); whereas insertions in *NPM1*, despite of correlating with TLR response, were evenly distributed between the two groups (4/11 vs. 10/28 patients).

### 2.6. TLR Responders Show Low mRNA Expression of Transcription-Related Proteins

We analyzed the global gene expression profile for the same patients as in the previous chapter but excluded partial responders to general TLR stimulation. The compared groups then contained 31 TLR responders and 9 TLR non-responders. We performed a significance analysis of microarrays (SAM) using a false discovery rate (FDR) of 5.0 (corresponding to *p* < 0.05) as cut-off value. The analysis identified 217 probes coding for 155 annotated proteins, and 62 probes coding for 45 annotated proteins that were overexpressed in the TLR non-responders and the TLR responders, respectively. Out of the 200 annotated proteins, 56 have been studied in or are associated with hematopoiesis, AML or related diseases (MDS, APL, Schwachman-Diamond Syndrome).

Functional annotation using DAVID 6.8 showed that the main enriched functional GO-term in the non-responders was “transcription, DNA templated”, which contained 38 genes (enrichment score: 4.89; *p* = 1.8 × 10^−7^). Gene functional classification identified 22 genes divided into three gene clusters and contained only four additional genes compared to the initial identified 38 genes. In the TLR-responders, on the other hand, only four proteins sharing the GO-term “cellular response to LPS” were enriched and contained the proteins CD86 and HAVCR2/TIM-3 that also were found to be differentially expressed in the unsupervised clustering (Figure 3). Details about the differentially expressed clusters and the 46 differentially encoded proteins are provided in Appendix A
Appendix A.

The 200 differentially expressed genes were also analyzed by STRING 11.0. For the TLR responders, a network of 14 proteins was identified that only was slightly enriched for the KEGG pathway “TNF signaling” (FDR = 0.030). This network again contained both CD86 and HAVCR2. For the non-responders, on the other hand, STRING identified a network of 41 proteins with most proteins belonging to the GO-terms “chromatin modification” (FDR = 8.8 × 10^−14^) and/or “cellular protein modification process” (FDR = 5.2 × 10^−6^). Main nodes consisted of E3 ubiquitin ligases and the chromatin remodeling complexes SWItch/Sucrose Non-Fermentable (SWI/SNF) and polycomb (Figure 4). The latter contains the epigenetic modifiers ASXL1 and EZH2, which recurrently are mutated in AML and then linked with an adverse prognosis. However, the mutational landscape did not differ between TLR responders or non-responders, neither with regard to chromatin modifiers nor any of the other assessed mutations (Appendix A).

Altogether, as many as 75 (thereof 16 putative) of the 155 annotated proteins in the TLR non-responders are associated with transcription or processes that appear either pre- or post-transcriptionally (Table 3). Furthermore, seven proteins that are related to the TLR, TNFα or NFκB pathways were found to be upregulated: among them two proteins (BABAM1 and OTUD5) that down-regulate TLR signaling, one protein (COMMD3) that down-regulates NFκB and two proteins (CXXC5 and ZBTB20) that enhance NFκB activation. On the other hand, in the TLR responders only one transcriptional protein (MLX) was overexpressed. Additionally, among the 45 annotated proteins are three (CASP1, HAVCR2 and ILK) proteins that activate/regulate the NFκB pathway and two proteins (MLKL and TNFRSF1B) that are associated with the TNFα pathway. Interestingly, while 13 E3 ubiquitin ligases were overexpressed in the TLR non-responders, three proteins (COPS7A, TNFRSF1B and YWHAB) that regulate or reduce the activity of E3 ubiquitin ligases were found to be overexpressed in the responders.

## 3. Discussion

As summarized in the Introduction, previous studies of TLRs in AML have mainly focused on cell proliferation and survival, and several of these studies were mainly based on the use of cell lines [12,13,14,15,16,17]. In the present study, we therefore focused on detection of functional receptors and mRNA expression of TLRs and proteins downstream in the signaling pathway. In addition, we investigated the communication with neighboring non-leukemic cells (i.e., mediator release) and associations between TLR responsiveness and genomic profiles. These aspects have not been addressed in the previous studies.

TLRs recognize pathogens either at the cell surface or at intracellular membranes, depending on the specific receptor. In this study, we concentrated on TLR1/2 and 5 that are localized to the plasma membrane, and on TLR7/8 that is situated at membranes of cytosolic compartments. Furthermore, TLR4 is mainly located to the plasma membrane but is also found intracellularly [8]. The reason for the selection of various receptors is that all TLRs except for TLR3 are expressed by normal hematopoietic cells. Signaling through TLR2 and TLR4 leads to commitment of primitive hematopoietic stem cells to the myeloid lineage [23]. Additionally, induction of TLR8 by R848 showed to induce monocytic differentiation in AML cells and reduced colony formation ability [15], which is regarded a negative prognostic factor [24]. Furthermore, induction of TLR1/2 by Pam3CSK4 in AML primary cells has shown to increase both apoptosis and myeloid blast differentiation [16]. Lastly, increased surface expression of TLR5 appears to be an inherent feature of AML [25]. TLRs mainly signal via the transcription factor NFκB to produce pro-inflammatory cytokines. Therefore, we regarded a high increase in the concentration of secreted mediators by AML blasts as proof that the stimulated TLR got activated.

The NFκB transcription factor contributes to maturation of monocytes and granulocytes during normal hematopoiesis [26], but is also a key survival factor in several types of cancer where it is usually upregulated and linked with chemoresistance [27]. In AML, NFκB upregulation is further associated with chemotactic migration of malignant cells, angiogenesis and extramedullary infiltration [27]. The pro-inflammatory cytokines IL-1β and TNFα, that are expressed upon signaling through TLR-NFκB, can also directly activate NFκB and thereby induce an autocrine positive feedback loop [26]. TLRs on the other hand are regarded as means of increasing the immunogenicity of AML cells and thus their susceptibility towards chemo- and/or immunotherapy [19]. This can be achieved by increasing the secretion of pro-inflammatory cytokines (especially IL-1β, IL-6, TNFα, IFNα and INFβ) as a result of TLR stimulation by single agonists [18,19], by a combination of TLR ligands [28,29,30], or by combination of agonists and TNFα/INFα [31,32]. As already mentioned, induction of both TLR1/2 and TLR8 further showed to induce blast differentiation in AML [15,16]. 

In this study, we saw a correlation between *NPM1* mutated cells responsiveness to TLR agonists, and experimental studies have shown that NPM1 can be a regulator of NFκB-dependent transcription, including the expression of certain cytokines [33]. The molecular interactions behind this effect are not known. The effects of cancer-associated mutations can be highly dependent on the biological context; this is for example true for many TP53 mutations where gain of function, selection of functions and loss of functions can be seen [34]. Thus, an association between TLR agonist responsiveness and NPM1 mutations is not surprising, but additional studies are needed to clarify the molecular mechanisms behind this correlation in primary AML cells.

Our AML cell populations contain a small contamination of normal cells, and light microscopy as well as flow cytometry show that this contamination corresponds to 0.5–5% (information from the biobank). For several reasons, we regard both the constitutive mediator release as well as the TLR agonist effects to be caused by the majority of leukemic cells and not by the minor contamination. First, constitutive release is not observed for most mediators when testing normal mononuclear cells at a much higher cell concentration incubated in medium alone, i.e., undetectable or only low CXCL8 levels are seen compared with the high levels observed for the AML cells in our present study [35] (Appendix A). Such a broad constitutive mediator release profile as described for our AML cell populations would thus not be expected for normal lymphocytes or monocytes. Second, the wide variation in constitutive release of many mediators among patients cannot be explained by a similar wide variation in the number of contaminating cells. Third, the TLR agonist effect was associated with the capacity of constitutive release, i.e., we observed a significant association between a generally high constitutive release (i.e., an AML cell characteristic) and responsiveness to the agonists. Finally, the wide variation among patients in the effect of TLR agonists is also difficult to explain by an effect on a similar and minor contamination of normal mononuclear cells. We therefore conclude that the TLR agonist effects observed in our present study are caused by an effect on the leukemic cells and not by an effect on a small contamination of normal mononuclear cells.

In this study, we observed that the TLR4 agonist LPS readily increased the median secretion of 12 mediators by at least 5-fold (Figure 1 and Appendix A), among them interleukins, chemokines, growth factors, matrix metalloproteases and TNFα. The TLR1/2 agonist Pam3CSK4 on the other hand, only increased the secretion of IL-6 and CCL2 to a higher extend. Because discrimination of patients into responders and non-responders after TLR targeting was most accurately achievable for the TLR1/2 agonist, this might explain why both patient survival and mRNA levels of TLR signaling components showed the highest association with response towards this particular agonist.

Even though several previous studies [29,30,31] have reported that TLR response upon receptor activation, both in terms of secretion of cytokines and cell differentiation, was linked with monocytic differentiation of the AML blasts (FAB M4/M5), we only found weak correlation between cell morphology and TLR1/2 and TLR5 responsiveness. However, we observed a correlation between TLR response, especially TLR1/2, and insertions in the *NPM1* gene, which is also associated with differentiation and absent expression of the CD34 stem cell marker [36].

Clustering 129 proteins belonging to the GO-term “TLR signaling”, we recognized 38 proteins to be differentially expressed at the mRNA level among 46 unselected patients of the cohort (Figure 3 and Table 2). The cluster divided the patients into two groups, where response to TLR1/2 ligation was the most differentially distributed factor. Thus, response to TLR activation is at least partly reflected by mRNA levels of members of the TLR signaling pathway.

In a previous study [37], we identified a group of patients with high constitutive cytokine secretion that was further characterized by overrepresentation of FAB M4/M5 cells and prolonged patient survival. Since high constitutive cytokine release might be a result of increased TLR pathway activation, as well as the previously mentioned autocrine feedback loops via NFκB, we investigated whether high mRNA levels of TLR signaling molecules correlated with constitutive cytokine secretion. Indeed, patient cells with high constitutive cytokine release appeared to have higher mRNA levels of TLR signaling molecules as well. Furthermore, FAB M4/M5 cells were also overrepresented in the patient group with high mRNA levels. However, the latter might partially be explained by the fact that several of the overexpressed proteins, especially CD86 [38] and S100A8/A9 [39], are correlated with monocytic differentiation. These observations are unlikely to be caused by a minority of contaminating monocytes because we showed in a previous study that genes included in the GO-term “monocytic differentiation” were only slightly differentially expressed among patients and were then clearly related to the FAB classification system but not mediator release [40]. Of note, we did not detect TLR2 to be differentially expressed within our cohort. High mRNA expression of this particular receptor was in a previous study linked both with shorter overall survival and less response towards treatment [13], whereas another study reported reduced TLR2 mRNA levels at remission, but could not link TLR2 expression at diagnosis with patient survival [12].

Like in the previous study, where we observed an association between constitutive mediator secretion and patient survival [37], also inducible mediator secretion by TLR1/2 seems to be part of a more favorable phenotype. The reason why the correlation was strongest between TLR1/2 and survival might be the aforementioned clearer differentiation into responders and non-responders towards this agonist. Furthermore, for three patients we could compare cells collected at the first time of diagnosis and at the time of relapse. While one patient (ID 16) was classified as responder at both time points, the other two patients (IDs 49 and 54) were classified as responders at the time of first diagnosis, but as non-responders during later disease progression. These observations further support our hypothesis that lack of TLR response is part of an adverse phenotype. A recent study including primary cells of 28 AML patients showed MAPK-dependent apoptosis and NFκB-mediated myeloid differentiation of the blasts after treatment with 10 ng/mL Pam3CSK4 (TLR1/2 agonist) [16]. Thus, the increase in mediator secretion upon TLR1/2 targeting, which we observed in our study and which was correlated with improved patient survival, might reflect that in these responding cells also NFκB-mediated differentiation was induced, i.e., increased mediator secretion and myeloid differentiation may be part of the same phenotype.

Finally, we analyzed the global gene expression profile where we compared TLR responders with non-responders. Several of the 200 annotated, differentially (i.e., *p* < 0.05) expressed genes coded for proteins that are relevant in AML. The discussion of the impact of single proteins would be outside the scope of this study, but we refer the reader to Appendix A
Appendix A for more detailed information about the 31 proteins that may be associated with AML prognosis. Interestingly, 16 out of 20 overexpressed genes in the TLR non-responders are linked with poor prognosis, as compared to five out of the 11 genes for the TLR responders. Functional analyses of differentially expressed genes for TLR responders showed only enrichment for proteins belonging to the GO-term “cellular response to LPS” and the KEGG pathway “TNF signaling”, i.e., signaling pathways downstream to especially TLR4. These proteins included the myeloid differentiation marker CD86 and the leukemia cell-specific HAVCR2/TIM-3. On the other hand, the TLR responders were especially characterized by low mRNA levels of pre- and post-transcriptional proteins.

The most enriched GO-term for TLR non-responders was “transcription, DNA templated”, which contained 38 proteins. In a previous study, we identified the same GO-term to be overexpressed for patient cells that showed low secretion of proteases and protease inhibitors [40]. A more thorough analysis showed that almost half of the overexpressed genes code for proteins involved in transcription, all the way from chromatin modification via transcription factors and regulators, to pre-mRNA splicing and post-transcriptional protein modifications (Table 3). STRING analysis (Figure 4) showed that especially proteins belonging to the chromatin modulating polycomb group, the SWI/SNF complex, and E3 ubiquitin ligases were enriched.

Polycomb group proteins maintain the repression of gene expression by histone modification. In hematopoiesis, they regulate repression of genes involved in differentiation, proliferation and maintenance of hematopoietic stem cells [41]. There exist two polycomb complexes, PRC1 and PRC2, and we found members of both complexes to be overexpressed in TLR non-responders. Of the five members that were overexpressed (Table 4), BMI1 and EZH2 upregulation is correlated with poor prognosis in AML [41,42,43,44]. *EZH2* and *ASXL1* are frequently mutated in AML; these mutations are linked with poor prognosis [45], but the frequencies of these mutations did not differ between TLR responders and non-responders (Appendix A).

The second overexpressed group was the SWI/SNF complex. This complex and the polycomb group oppose each other in regulation of gene expression. While polycomb complexes modify histone N-terminals, SWI/SNF utilizes ATP to disrupt the interaction between DNA and nucleosomes, move the nucleosomes along the DNA, and remove them in order to allow transcription [46]. SWI/SNF members interact with several hematopoietic transcription factors, but gene mutations in these proteins are not associated with AML [47]. Still, at least specific SWI/SNF members appear to contribute to blast proliferation and survival [47]. Of the four overexpressed proteins in the TLR non-responders, ARID2/BAF200 contributes to stem cell maintenance and suppression of leukemogenesis in t(9;11) transgenic mice [48].

Finally, 13 E3 ubiquitin ligases were overexpressed by TLR non-responders. The ubiquitin proteasome process is mediated by three types of enzymes: E1 ubiquitin- activating enzymes, E2 ubiquitin-conjugating enzymes and E3 ubiquitin ligases. Of these, the latter emerge as important for cancer development through their inactivation of p53 [49]. Deregulation of E3 ubiquitin ligases is also linked with hematological malignancies [50]. Of note, also the PRC2 member EZH2 and the NFκB pathway are regulated by E3 ubiquitin ligases [50,51]. To summarize, TLR non-responders show overexpression of several protein groups that are important for transcriptional regulation and associated with leukemia development, progression and adverse prognosis in AML.

In a previous study we described that TLRs and NFκB signaling in mesenchymal stem cells are important for the AML-supporting crosstalk between leukemic cells and the stromal cells [52]. Our present results show that TLR ligation, and thereby NFκB signaling, seems to be important for chemosensitivity in human AML. Thus, TLR/NFκB targeting may be a possible therapeutic strategy especially for the patients where this targeting may have both direct and indirect (i.e., mediated through the neighboring stromal cells) antileukemic effects.

## 4. Materials and Methods

### 4.1. Patients and Cell Preparations

The study was approved by the local Ethics Committee (Regional Ethics Committee III, University of Bergen, REK 2017-305, Bergen, Norway) and samples were collected after written informed consent. AML blasts from peripheral blood were derived from 83 consecutive patients with a high number and/or percentage of circulating leukemic cells (39 females and 44 males; median age 66 years with range 18–92 years). A majority of 55 patients had de novo AML whereas 24 patients had AML secondary to chronic myeloproliferative neoplasia, myelodysplastic syndromes or previous chemotherapy, and seven patients had relapsed AML (Table 4). Additionally, three of the patients were enrolled twice in the study: both at first diagnosis and at relapse of the disease. Thirty-eight of the patients, including two of the patients with samples gathered at dual time-points, received potentially curative treatment including induction therapy based on cytarabine plus an anthracycline followed by intensive consolidation treatment.

AML cells were isolated from peripheral blood by density gradient separation (Lymphoprep; Axis-Shield, Oslo, Norway; specific density 1.077 g/mL) and contained at least 90% blasts. The cells were stored in liquid nitrogen until use [53]. As a part of the quality control of cells included in the biobank, it has been documented that variation among AML cell samples in proliferative responsiveness, survival during in vitro culture and constitutive cytokine release is not associated with the storage time in liquid nitrogen.

### 4.2. Reagents

The following TLR-agonists (InvivoGen; San Diego, CA, USA, if not stated otherwise) were used at following concentrations: (i) 10 ng/mL of the TLR1/2 heterodimer agonist Pam3CSK4, a synthetic triacetylated lipopeptide, (ii) 10 ng/mL of the TLR4 agonist LPS isolated from *Escherichia coli 0111:B4* (Sigma Aldrich; St. Louis, MO, USA), (iii) 100 ng/mL of the TLR5 agonist flagellin isolated from *Salmonella typhimurium*, and (iv) 100 ng/mL of the dual TLR7/8 agonist resiquimod (R848). These compounds were tested on ten unselected AML patients at concentrations of 0.01, 0.1 and 1.0 µg/mL (Pam3CSK4, LPS and flagellin) or respectively of 0.1, 1.0 and 10 µg/mL (R848). At the selected concentrations, the agonists significantly increased the AML blasts’ secretion of the cytokines CCL2, CCL3, CCL4, CXCL8 and IL-6, while the proliferation capacity of the cells was not compromised (Appendix A).

### 4.3. Analysis of Mediator Levels in Cell Culture Supernatants

Cell supernatants with and without addition of single TLR agonists were collected from AML cells cultured for 48 h in Stem Span SFEM medium (StemCell Technologies; Vancouver, BC, Canada) in 48-well culture plates (Becton Dickinson; Franklin Lakes, NJ, USA) at a concentration of 1 × 10^6^ cells per mL. The samples were stored at −80 °C prior to analysis. The cell culture supernatants were analysed by Luminex analyses (R&D Systems; Minnesota, MN, USA). Secretion levels for following 19 soluble mediators were measured: (i) The chemokines CCL2-5, CXCL1/5/8/10; (ii) the interleukins IL-1β, IL-1RA and IL-6; (iii) the growth factors G-CSF (16 patients)/granulocyte-macrophage colony stimulating factor (GM-CSF; 67 patients) and hepatocyte growth factor (HGF); (iv) the matrix metalloproteases MMP-1/2/9; (v) the protease inhibitors cystatin C and serpin E1; and (vi) TNFα.

### 4.4. RNA Preparation and Analysis of Global Gene Expression

RNA preparation, labeling, and microarray hybridization have been described in detail previously [54]. Briefly, the gene expression profiles were analyzed using the Illumina iScan Reader (Illumina Inc., San Diego, CA, USA) for fluorescent detection of biotin-labeled complementary RNA (cRNA). The latter was quality-controlled by both NanoDrop and an Agilent 2100 Bioanalyzer (Agilent Technologies, Santa Clara, CA, USA). Finally, 750 ng biotin-labeled cRNA was hybridized to the HumanHT-12 V4 Expression BeadChip (Illumina Inc.), which targets 47,231 probes. The microarray data were quantile normalized prior to analysis in J-Express 2012 (MolMine AS, Bergen, Norway) [55].

### 4.5. Statistical and Bioinformatical Analyses

The statistical analyses were performed with the IBM Statistical Package for the Social Sciences (SPSS) v25 (Chicago, IL, USA) and with GraphPad Prism 5 (San Diego, CA, USA). The χ^2^ test was used to analyze categorized data. Furthermore, Kaplan Meier analysis and log-rank test in addition to Cox regression for uni- and multivariate analyses were used for patient survival statistics. In general, *p*-values < 0.05 were regarded as statistically significant.

For hierarchical clustering of the mediator release and mRNA expression levels, all values were median normalized, and log transformed prior to clustering using J-Express. Complete linkage and Euclidean correlation were used as the linkage and distance measurement, respectively. For analysis of the relationships between differentially expressed genes/proteins for TLR responding and non-responding patients, DAVID 8.6 [56,57] and STRING 11.0 [58] were utilized.

## 5. Conclusions

The major difference between our 83 AML patients was related to TLR1/2 responsiveness. The responders showed prolonged survival, and the responding cells showed frequent *NPM1* mutations, high constitutive mediator release, relatively higher mRNA expression of TLR pathway molecules and lower expression of transcriptional regulators. Caution has to be taken, because prolonged activation of TLR signaling and NFκB may be linked with malignant transformation [50]. In this respect, it is of interest that we found that also negative regulators of the TLR signaling pathway and NFκB inhibitors were upregulated in TLR responders, indicating that the balance between positive and negative regulators is important.

In two previous studies we identified patient subsets with increased constitutive cytokine and protease secretion, prolonged patient survival [37], and also mRNA overexpression of TLR signaling molecules such as S100A8 and A9 [40]. In the present study, we further observe a link between TLR ligation, high mediator secretion, mRNA overexpression of TLR signaling molecules and increased survival. Future studies should try to identify the molecular mechanisms behind these association and the effects TLR ligation has on cellular regulation of proliferation, differentiation and survival/chemosensitivity. Crosstalk between TLR-initiated signaling and other pathways (e.g., the MAPK-Erk and PI3K-Akt-mTOR pathways) may then be of importance [59].

## Figures and Tables

**Figure 1 cancers-11-00973-f001:**
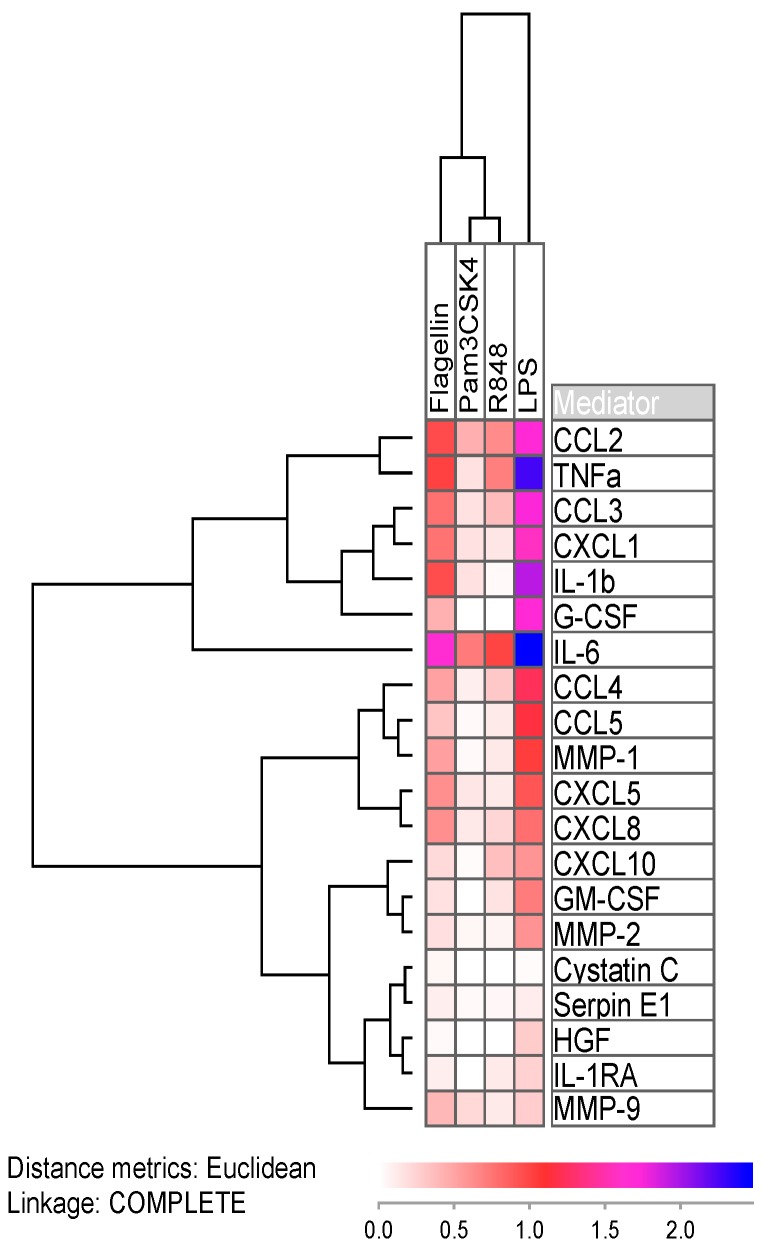
Log-transformed median mediator secretion increase in AML cells after incubation in TLR single agonists for 48 h. For each of the 20 soluble mediators (right part of the figure) we estimated and log-transformed the median effect of the four various TLR agonists (see top of the figure), i.e., the figure is based on the median results for all patients for each agonist/soluble mediator combination. Median values were used in order not to over-estimate the change from low basal levels, and at the same time not to under-estimate the change from basal levels that were close to detection limit of the assay. IL-6 was the most readily upregulated cytokine with medians of 5-fold (Pam3CSK4), 10-fold (R848), 45-fold (flagellin) and 300-fold (LPS) concentration increases. The secretion of the protease inhibitors cystatin C and serpin E1 on the other hand were not altered by TLR ligands.

**Figure 2 cancers-11-00973-f002:**
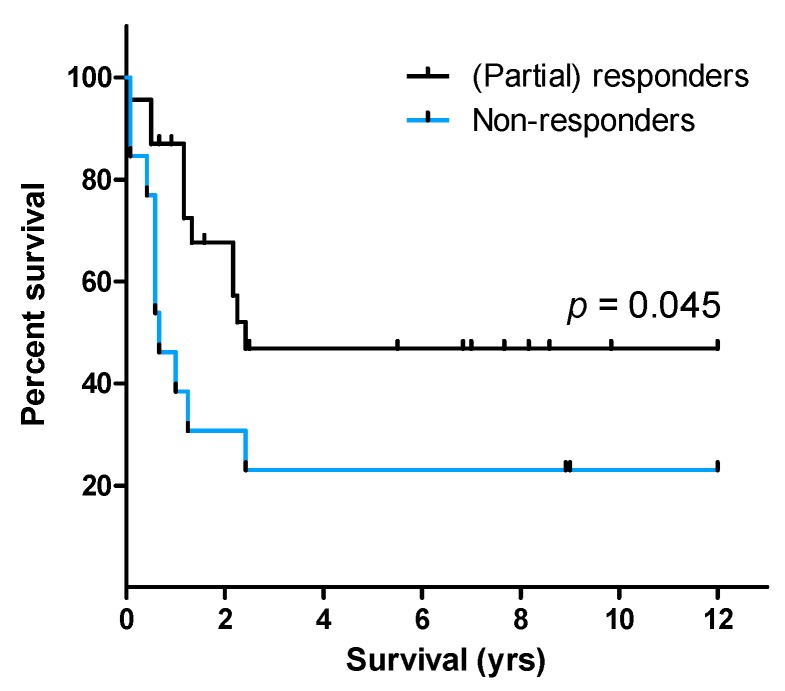
Patient survival dependent on response towards TLR agonists. Lack of response towards TLR1/2 targeting by Pam3CSK4 was significantly linked with poor outcome (log-rank test).

**Figure 3 cancers-11-00973-f003:**
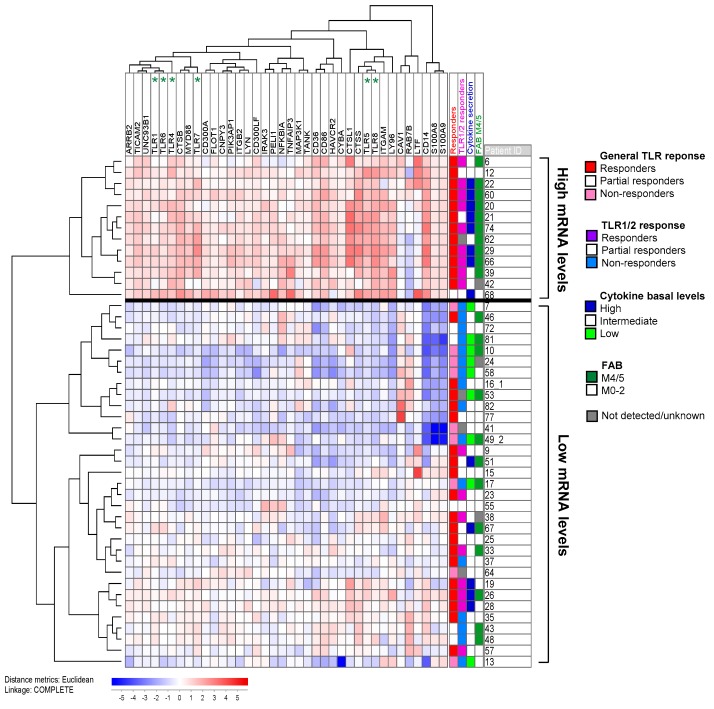
A cluster containing the 38 out of 129 genes belonging to the GO-term “TLR signaling” (based on the AmiGO database) that were most differentially expressed among the 46 unselected patients in our study. The remaining genes, including TLR2, showed only minor variations among patients and did not differ significantly between the two identified clusters. Six of the ten TLR receptors showed differential expression (i.e., were included among the 38 genes) and are indicated at the top of the figure. All values were median normalized and log_2_-transformed, thus red and blue color indicate values above and below the median, respectively. The most differentially expressed genes were the TLR4 co-receptor CD14, and the TLR4 ligands S100A8 and A9 (on the far right). Patient distribution in the cluster was correlated with response upon TLR induction, especially by the TLR1/2 ligand Pam3CSK4, basal cytokine secretion levels and monocytic cell differentiation according to the FAB system. The numbers behind the patient ID correspond to the de novo sample (1) and the relapse sample (2) for patients with cells sampled at dual time points. All cells included in this analysis were gradient-separated and cryopreserved primary AML cell as described in Section 4.1. mRNA was prepared immediately after thawing without any kind of cell stimulation/incubation.

**Figure 4 cancers-11-00973-f004:**
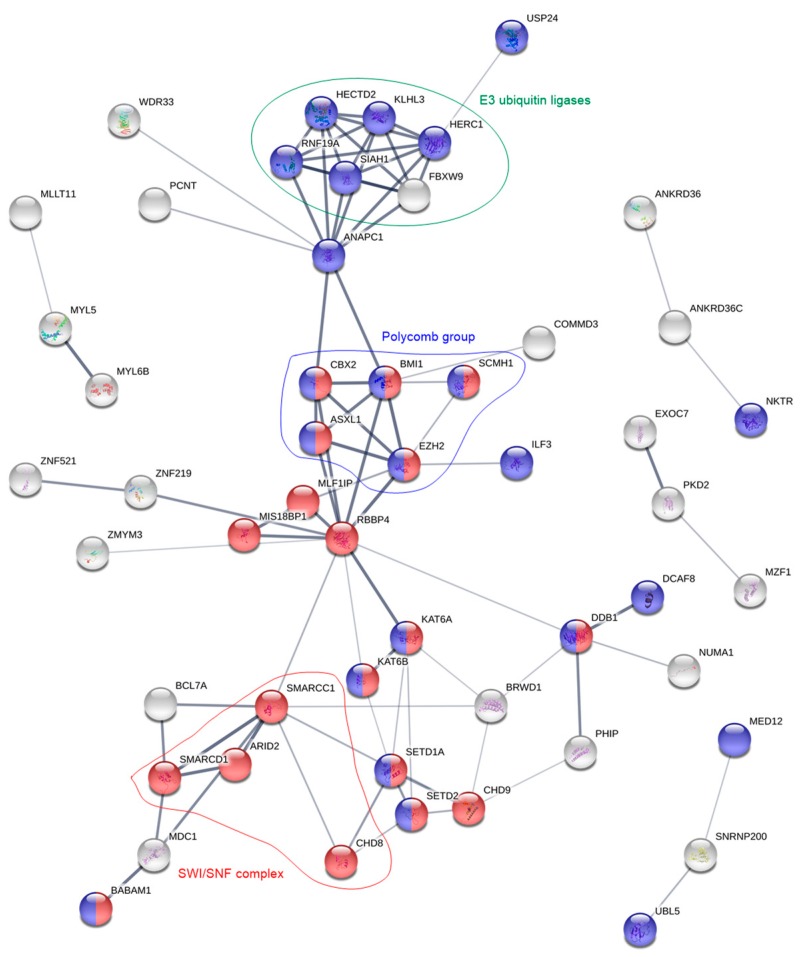
Protein network analysis of the overexpressed genes in the TLR non-responsive patient cells. For clarity, the figure only contains proteins that are interrelated with at least two other proteins. Red and blue color correspond to the GO-terms “chromatin modification” and “cellular protein modification process”, respectively. Groups of proteins with identical functions (E3 ubiquitin ligases, green) or that are part of the same protein complexes (polycomb, blue; SWI/SNF, red) are highlighted.

**Table 1 cancers-11-00973-t001:** Multivariate analysis with adjusted hazard ratio for parameters associated with AML prognosis.

Variable	Adj. HR	95% CI	*p*-Value
TLR1/2 (non-responder) ^1^	1.86	1.07–3.24	0.029
Age (≥60 years)	1.57	0.86–2.88	0.145
Etiology (secondary)	3.05	0.62–14.94	0.170
Cytogenetics (intermediate/adverse)	0.56	0.14–2.29	0.422
*Flt3*-ITD	2.53	0.80–7.95	0.113
*NPM1*-insertion	0.81	0.21–3.24	0.758

^1^ Reference value. HR: hazard ratio; CI: confidence interval.

**Table 2 cancers-11-00973-t002:** Overview of the 38 differentially expressed mRNA levels of proteins belonging to the GO-term “TLR signaling” and their function in the TLR pathway. The three proteins in bold were found to be overexpressed in the patient subset with reduced response towards TLR targeting.

Function	Proteins
Chaperone, translocation	CNPY3/PRAT4A, UNC93B1
TLR cleavage	CTSB, CTSS
Agonists	S100A8, S100A9
Receptors	TLR1, TLR4, TLR5, TLR6, TLR7, TLR8
Co-receptors	CD14, CD36, CD86, LY96/MD2
Adaptors, co-factors	CTSL1, MYD88, PIK3AP1/BCAP, TICAM2/TRAM
Downstream targets	CYBA, MAP3K1/MEKK1, PELI1
Positive regulators/feedback pathways	**CAV1**, FLOT1, **LTF**
Negative regulators/feedback pathways	ARRB2, CD300A, CD300LF, HAVCR2/TIM-3, IRAK3/IRAK-M, ITGAM/CD11b, ITGB2, LYN, NFKBIA, PIK3AP1/BCAP, **RAB7B**, TANK, TNFAIP3

**Table 3 cancers-11-00973-t003:** Overview over the 75 transcription-associated proteins that were found to be overexpressed in TLR non-responding patient cells. Proteins in bold have been studied or identified as overexpressed in AML.

Protein Task	Protein Group	Protein Names	Protein Names (Putative Task)
**Pre-Transcription**			
Chromatin remodeling			
	Polycomb	**ASXL1**, **BMI1**, **CBX2**, **EZH2**, **SCMH1**	
	SWI/SNF	**ARID2**, **CHD8**, **SMARCC1**, SMARCD1	**CHD9**
	WD40-repeats	BRWD1, FBXW9, **RBBP4**, WDR33	WDR42A
Chromatin modification		SETD1A, **ZMYM3**	
Chromatin binding		**MED12**, **MYST3**, **MYST4**, **RCOR3**	C14orf106
**Transcription**			
TFs	Activation and/or regulation	**DACH1**, ELF2, FOXJ3, ILF3, **MYST4**, MZF1, NFX1, RFX7, **SCMH1**, **TP53INP1**, **ZBTB20**, **ZNF395**, **ZNF521**, **ZNF548**, ZNF880	CCDCC171, KANSL3, **RCOR3**, UBN2, ZNF521, ZNF669, ZNF700, ZNF763, ZNF789
Transcription repressors		TRIM33, ZNF219, ZNF431, **ZNF521**	
**Post-Transcription**			
Pre-mRNA splicing		SFRS8, SFRS14, **SNRNP200**, **TCF3**, WDR33, YTHDC1	EXOC7, **PSIP1**
mRNA degradation		DCP1A	
Ubiquitination			
	E3 ubiquitin ligases	CCNB1IP1, DDB1, FBXO46, FBXW9, HECTD2, HECTD4, HERC1, MAGEL2, MARCH5, RNF19A, RNF144, RNF144A, **SIAH1**	
	Other tasks with ubiquitination	**CUL9**, KLHL	COMMD3, WDR42A
	De-ubiquitination	OTUD5, USP4	BABAM1

**Table 4 cancers-11-00973-t004:** Biological and clinical characteristics of the 83 AML patients included in the study.

Patient Characteristics, Cell Morphology		Disease Etiology, Cell Morphology		Cell Genetics	
*Age*		*De novo AML* ^1^	55	*Cytogenetics*	
Median (yrs)	66			Favorable	7
Range (yrs)	18–92	*Secondary AML*		Intermediate	14
		MDS	12	Normal	38
*Gender*		CMML	4	Adverse	14
Females	39	CML	1	n.d.	10
Males	44	CLL	1		
		MF	4	*Flt3 mutations* ^3^	
*FAB classification*		PV	1	ITD	26
M0	5	Chemotherapy	1	Wild-type	41
M1	20			n.d.	16
M2	14	*AML relapse* ^2^	7		
M4	21			*NPM1 mutations*	
M5	17	*CD34 receptor*		Mutated	22
n.d.	6	Negative (≤20%)	25	Wild-type	45
		Positive (>20%)	50	n.d.	16
		n.d.	8		

^1^ Patients that were enrolled twice, at first diagnosis and relapse, are listed with their de novo characteristics. ^2^ Three of the patients had relapse of secondary AML. ^3^ One patient in each group has a point mutation at D835. n.d.: not determined. Abbreviations: MDS: myelodysplastic syndrome; CMML: chronic myelomonocytic leukemia; CML: chronic myeloid leukemia; CLL: chronic lymphocytic leukemia; MF: myelofibrosis; PV: polycytemia vera; ITD: internal tandem repeat; n.d.: not determined.

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
