# Peer review of "Functional Toll-Like Receptors (TLRs) Are Expressed by a Majority of Primary Human Acute Myeloid Leukemia Cells and Inducibility of the TLR Signaling Pathway Is Associated with a More Favorable Phenotype"

_cancers, 2019, doi:10.3390/cancers11070973_

Reviewer 1 Report

Congratulations for really good work. No comments except that the authors should discuss more about the relation between NMP1 mut AML and TLR1/2 signalling.  

Author Response

Congratulations for really good work. No comments except that the authors should discuss more about the relation between NMP1 mut AML and TLR1/2 signalling. 

We are very grateful for the general comment. We have included a new chapter in the Discussion section of our article to comment on the probable correlation between NPM1 mut and TLR signaling (lines 337-344).

Reviewer 2 Report

The secretion of 20 different cytokines in response to stimulation with four toll-like receptor (TLR) ligands was monitored in primary tumor cells from 83 acute myeloid leukemia (AML) patients.  The obtained data were correlated with patient survival and subjected to protein network analysis.

1)            The merit of this manuscript is the comprehensive monitoring of cytokine secretion response to TLR stimulation in a high number of AML cell samples. The description of this effort mostly by median values of all patients leaves the value of this data resource largely untapped. The thresholds for the TLR response status are defined, but quite complicated and not clear-cut, since they combine sometimes different stimuli, and always several cytokines. One also has to bear in mind that TLR stimulation, unlike for instance that of receptor tyrosine kinases, is pleiotropic and partly overlapping between the ligands used.  

2)            It is incorrect to correlate TLR response with patient survival without taking into account a patient stratification according to treatment and relapse status. In my opinion patient survival depends too much on treatment eligibility (e.g. availability of matched stem cell donors) and occurrence of infections to allow a meaningful correlation analysis. The authors reported a stronger but similar correlation of the capacity of in vitro cell proliferation in ref. 16, for which these concerns potentially also apply. In the present manuscript, however, it is not clearly reported, which samples were from treated or untreated patients and what the individual survival was. The authors themselves alert of the critically low number of samples from treated patients in the responder group. The formally significant difference reported here definitely is due to chance and not to a relevant correlation between TLR1/2 response and patient survival. Thus faulty correlation analysis leads to a potentially misleading conclusion.

3)            The gene expression analysis presented in Fig. 3 is probably meant to provide patient data at the individual level to complement the TLR stimulation data shown in aggregate in Fig. 1. It is, however, not clear, how these gene expression data were obtained from the patient samples. The restriction on genes related to TLR signaling and the uncertainty regarding the origin of the expression data also limit the value of the protein network analysis.

4)              Descriptions in the text have to be improved. Often details are just interspersed in parentheses into the description of something else. Particularly at the beginning, inexact terminology is used, for instance “targeting” instead of stimulation, “attribution” instead of addition, “mediator” instead of cytokine etc.

Author Response

The secretion of 20 different cytokines in response to stimulation with four toll-like receptor (TLR) ligands was monitored in primary tumor cells from 83 acute myeloid leukemia (AML) patients.  The obtained data were correlated with patient survival and subjected to protein network analysis.

1. The merit of this manuscript is the comprehensive monitoring of cytokine secretion response to TLR stimulation in a high number of AML cell samples. The description of this effort mostly by median values of all patients leaves the value of this data resource largely untapped. The thresholds for the TLR response status are defined, but quite complicated and not clear-cut, since they combine sometimes different stimuli, and always several cytokines. One also has to bear in mind that TLR stimulation, unlike for instance that of receptor tyrosine kinases, is pleiotropic and partly overlapping between the ligands used.  

As can be seen from the new chapter added to the Introduction section several TLRs can be expressed by AML cells, and as stated by this reviewer the TLR stimulation is pleiotropic and partly overlapping between the ligands. In this context we wanted to identify patient subsets and subclassify patients based on whether they have a broad and strong TLR response versus patients with a weaker or absent TLR response. For these reasons we had to define responsiveness based on several agonists and based on the investigation of a panel including several soluble mediators. When using this scientific approach, the definition of responsiveness must be relatively complicated when one has to take into account both the cytokine levels, the number of cytokines influenced, and the number of agonists investigated.

We have rewritten and extended this part of the article (now section 2.3) to explain this more clearly (lines 111-121). We agree that our definitions are complicated, and for this reason one should also be very careful with the interpretation of our data set. Please see the comment 4.2 made by reviewer 4. We did our analysis as we had decided before the study. It may have been possible to further go into detailed analyses of single mediators or patient subsets, but our intention with the data has been to be careful both in the analyses and the interpretation. We definitely agree that the TLR response is pleiotropic as stated by the reviewer, and we therefore want to be consequent throughout the study and focus on mediator profiles and not on single mediators. Furthermore, analysis with regard to additional patient subsets may also be a possibility, but even though we included more patients than most other experimental AML studies one should be careful with regard to analysis of additional and possibly small patient subsets. Finally, we presented uncorrected p-values. If we do extensive analysis we get into the problem of multiple comparisons and the question of correcting p-values.

To conclude, we want to keep to our original design of the study and the decisions made before we started the analysis, i.e. TLR agonist responses with a readout (cytokine release) that is to a large part regulated by events shortly downstream to the receptor with modulation of NFκB. For example, proliferation is more complex and will probably depend both on direct effect and indirect effects mediated by autocrine signaling (inhibition/enhancement) of various cytokines.

We hope this can be accepted. We also added a supplementary figure to show the AML cell cytokine distribution for the cytokines with the largest concentration ranges.

2. It is incorrect to correlate TLR response with patient survival without taking into account a patient stratification according to treatment and relapse status. In my opinion patient survival depends too much on treatment eligibility (e.g. availability of matched stem cell donors) and occurrence of infections to allow a meaningful correlation analysis. The authors reported a stronger but similar correlation of the capacity of in vitro cell proliferation in ref. 16, for which these concerns potentially also apply. In the present manuscript, however, it is not clearly reported, which samples were from treated or untreated patients and what the individual survival was. The authors themselves alert of the critically low number of samples from treated patients in the responder group. The formally significant difference reported here definitely is due to chance and not to a relevant correlation between TLR1/2 response and patient survival. Thus faulty correlation analysis leads to a potentially misleading conclusion.

We agree with the reviewer that our survival analysis has to be described more in detail. We also agree that when using survival analysis to evaluate a new therapeutic one has to evaluate both the efficiency of the treatment (in AML: the antileukemic effect) and compare this with the toxicity or adverse effects that will be influenced both by comorbidity and the risk and nature of the toxicity of the pharmacological agent. This is to answer the question whether benefit of the treatment outweighs the risk of treatment-related toxicity and in AML even treatment-related mortality. Usually event-free and/or overall survival is studied.

Our intention with a survival analysis is to investigate whether an AML cell characteristic (i.e. responsiveness to TLR agonists) is associated with clinically relevant chemosensitivity. We only investigate the AML cells, i.e. only the pharmacological effect on the AML cells (i.e. the chemosensitivity) is relevant with regard to our present study, whereas the question of comorbidity/toxicity on nonleukemic cells or other tissues become less important as long as we only characterize AML cells and want to investigate whether the agonistic responsiveness is important for the AML cell chemosensitivity.

To investigate survival one has to study a group that has received the treatment according to common guidelines. We studied a subset of our patient cohort that had received intensive treatment with the intention to achieve complete remission and cure after consolidation. We only studied overall survival. Our cohort included the following patients:

-        A total of 41 patients with newly diagnosed AML started intensive chemotherapy.

-        Two patients were lost from follow-up.

-        One patient died from toxicity early after induction treatment before regeneration or relapse.

-        We included 38 patients in our survival analysis, i.e. excluding the above-mentioned patients.

-        Thus, we have a group who becomes long-term survivors and another group who dies from chemoresistant disease, and we can evaluate the associations between agonist responsiveness and AML cell chemosensitivity without any disturbances from comorbidity.

This information is now given in the results section (lines 150-160). We hope our explanation for doing an overall survival analysis to investigate a possible association between AML cell responsiveness to TLR agonists and clinical chemosensitivity can be accepted, and we would also emphasize that only three patients were left out from our unselected subset of young and fit patients to be able to perform such an analysis.

3. The gene expression analysis presented in Fig. 3 is probably meant to provide patient data at the individual level to complement the TLR stimulation data shown in aggregate in Fig. 1. It is, however, not clear, how these gene expression data were obtained from the patient samples. The restriction on genes related to TLR signaling and the uncertainty regarding the origin of the expression data also limit the value of the protein network analysis.

We agree with the reviewer that some additional information was needed. A short introduction to result section 2.5 was therefore added (lines 196-198). Proteins related to TLR and TLR signaling were obtained from http://geneontology.org/ and the patient samples were obtained from microarray as now stated.

We restricted the gene expression analysis to this protein group to see if mRNA levels correlate with response to TLR agonists and thus patient distribution. Unlike the mRNA analysis in section 2.6, where we investigated whether there are additional differences between responders and non-responders, i.e. not directly correlated to TLR signaling.

4. Descriptions in the text have to be improved. Often details are just interspersed in parentheses into the description of something else. Particularly at the beginning, inexact terminology is used, for instance “targeting” instead of stimulation, “attribution” instead of addition, “mediator” instead of cytokine etc.

We have addressed this issue to the best of our efforts. We also tried to be more consistent in our use of “cytokines” and “mediators”. Usually, one talks about pro-inflammatory cytokines that are upregulated upon TLR stimulation. However, we also saw mediators besides cytokines (i.e. growth factors and MMPs) to be upregulated. Therefore, we kept both terms in the text.

Reviewer 3 Report

Brenner and Bruserud describe a new cohort of AML patient samples analyzed for their transcriptional patterns related to TLR signaling.  This is an interesting question and a valuable resource using primary patient samples. However, as presented, the data is difficult to access and some key controls are not presented.

1) The introduction and discussion are missing some key references that specifically address similar topics, some examples include:

Rybka, Leukemia Research, 2015 – TLR2, 4 and 9 expression and prognosis.

Ramzi, Int J Hematol Oncol Stem Cell Res. – association between TLR2 and 4

It may also be helpful to expand on several topics in the introduction, as this is a broader audience that may not be familiar with the TLR specific nuances for AML.

Perhaps one (or more) of these reviews may be useful to address/reference for the reader:

Monlish, Front. Immunol. 2016

Sellar, Nature, 2018

Agarwal, Leukemia, 2015

2) The results jump directly into TLR responses. Additional details are needed here to 1) describe the patient cohort used (this is referenced in Table S2) but much later. The primary patient sample cohort is a strength of this study and should be made clear.

3) There are several instances where additional methodological details are necessary to clarify how these studies were performed, including:

            a) Figure 1 – which subset of samples were used here (some/all) and how was this data obtained?

            b) Was TLR expression in these/all samples verified?

            c) Villamon, Cancer Cell International, 2018 demonstrate that 0.1ug/mL (100ng/mL) of R848 has a growth inhibition effect on AML cell lines; yet Figure S2 does not show this effect on the overall grouping, although significant variation between samples is apparent with all.

            d) There is a disconnect between the conclusion regarding patient age – line 126 implies this is a significant factor, while line 84 states the opposite.

            e) For the transcriptomics analysis presented in Figure 3, it is unclear which samples were used and what state they were used at. Are these TLR stimulated samples? If so, with which? If not, are these cultured? Viably frozen? Sorted for live blast cells? Or total pellets? Were all samples obtained, processed and treated in the same manner and same timeframe (i.e. sample storage between processing steps)?

            f) NPM1 mutations are referenced in several instances. How were these assessed? Is the data presented elsewhere?

            g) Line 265 references morphological analysis. How was this analyzed? Is the data presented elsewhere?

            h) Line 268 discusses clustering of 129 proteins for further analysis.  Where did this list come from?

            i) Paragraph starting line 282 discusses purity of the samples to >90% blasts. Is this data shown elsewhere? How was this determined?

4) There are also several instances where comparison to previous literature (or description that this is an original finding) would put this data in perspective, for example in Figure 1 – how do these TLR responses compare to other studies of AML cells/AML cell lines/other primary samples?       

5) The result descriptions are difficult to follow and could use clarification/simplification (fewer compound sentences would help here).

a) One example is the paragraph starting on line 73 – the description of responders is needed and all relevant data is here, but it is hard to follow and could perhaps in a simpler format here and more of this information moved to the methods.  

b) There are some grammatical errors that can be misleading and should be corrected.

c) and a few additional typographical errors (one example, line 180, referencing a previous chapter.

6) The conclusions are ambiguous and overstretched in places, some points of confusion:

            a) The title emphasizes TLR1/2-NFkB signaling, however, the experimental data also includes stimulation for TLR4, 5 and 7/8 as well, which is appropriate for controls (I agree here with their inclusion), but which are given equal weight in the discussion and seem to have equal responses in the data presented (i.e. Figure 3). The discussion also focuses these other TLRs as well, so the emphasis on just TLR1/2 in the title seems limiting.

            b) The conclusions stating that a number of factors are not correlated with TLR responses seem overstated given the patient cohort size (and lack of significant numbers with in specific cohorts).

            c) The link of NFkB seems tenuous. More clarification of how the authors see NFkB regulation, or specific NFkB pathway members that are differentially regulated in this data set could be highlighted.

Author Response

Brenner and Bruserud describe a new cohort of AML patient samples analyzed for their transcriptional patterns related to TLR signaling.  This is an interesting question and a valuable resource using primary patient samples. However, as presented, the data is difficult to access and some key controls are not presented.

1. The introduction and discussion are missing some key references that specifically address similar topics, some examples include:

Rybka, Leukemia Research, 2015 – TLR2, 4 and 9 expression and prognosis.

Ramzi, Int J Hematol Oncol Stem Cell Res. – association between TLR2 and 4

It may also be helpful to expand on several topics in the introduction, as this is a broader audience that may not be familiar with the TLR specific nuances for AML.

Perhaps one (or more) of these reviews may be useful to address/reference for the reader:

Monlish, Front. Immunol. 2016

Sellar, Nature, 2018

Agarwal, Leukemia, 2015

We are grateful for the general comment. We added a new section to both the introduction (lines 45-66) and the discussion (lines 299-305), and an additional comment on TLR2 (lines 390-394). All the references plus the one mentioned in 3c except for Sellar 2018 are now included.

2. The results jump directly into TLR responses. Additional details are needed here to 1) describe the patient cohort used (this is referenced in Table S2) but much later. The primary patient sample cohort is a strength of this study and should be made clear.

A new chapter is now added as the first chapter in the Results section (lines 71-86).

3. There are several instances where additional methodological details are necessary to clarify how these studies were performed, including:

            a) Figure 1 – which subset of samples were used here (some/all) and how was this data obtained? 

We added information to the figure in the text (lines 95-98). All patient samples were used and show the concentrations as fraction of cells with agonists/cells alone. As several concentrations are below detection limit in the control and some stimulated levels are above the detection limit of the assay, we decided to use the median values per single mediator in order to give a balanced picture and not over-interpret the results.

            b) Was TLR expression in these/all samples verified?

We did not verify the TLR expression at the protein levels, but mRNA expression by microarray was studied for 46 unselected patients. TLR mRNA expression was analyzed and correlated with TLR response as the targeted TLR4, TLR5, TLR8 and the partners of TLR2 – TLR1 and TLR6 – were found to be upregulated among the TLR responding patients (section 2.5, lines 220-222).

            c) Villamon, Cancer Cell International, 2018 demonstrate that 0.1ug/mL (100ng/mL) of R848 has a growth inhibition effect on AML cell lines; yet Figure S2 does not show this effect on the overall grouping, although significant variation between samples is apparent with all.

Whether receptor ligation has growth enhancing or inhibitory effects is in confliction from the available published results. This is now clearly stated in the new chapter in the Introduction section, and our own present results clearly show that individual patients are heterogeneous.

We have not included a detailed discussion of the use of cell lines in the studies of AML; the reason is that we do not include AML cell lines in our study. However, we want to emphasize that cell lines are very different from primary cell with regard to: atypical growth pattern, indefinite growth without or with a limited number of growth factors, and extensive karyotypic abnormalities. Even if we have not included such a detailed discussion in the article, we state in a new chapter in the Introduction that results are conflicting and may depend on the experimental models used.

Furthermore, we interpret the results obtained by both Villamon et al. and Ignatz-Hoover et al. to be in accordance with our proliferation results. Villamon et al. show that 10 µg/mL of imiquimod (R837) substantially reduces the proliferation in all 10 cell lines. That fits with our observation (now Figure S3), where 10 µg/mL of imiquimod (R837) strongly inhibited proliferation in our 10 test patients. We added the citation as our findings support the results obtained on cell lines.

Both Villamon et al. and Ignatz-Hoover et al. further show that 25 µg/mL of resiquimod (R848) moderately reduces the proliferation of approximately half the tested cell lines. We only tested 10 µg/mL as the highest dose in our test setup, but we observed a trend towards reduced proliferation at this concentration (Figure S3), that likely will increase with further concentration increase. The lack of effect on proliferation at a concentration of 0.1 µg/mL might therefore be that R848 reduces proliferation first at a threshold value.

All in all, there seems to be a good correlation between proliferation results obtained from experiments on AML cell lines and primary cells with regard to these two agonists.

            d) There is a disconnect between the conclusion regarding patient age – line 126 implies this is a significant factor, while line 84 states the opposite.

The reviewer supposedly means the discrepancy between age being a prognostic factor for patient survival for response towards TLR1/2 targeting in Figure 2, and for age not being significantly associated with TLR1/2 response overall. The reason for that is that in the first case (former line 84) we compared age distribution for the overall cohort of 78 patients with respect to TLR1/2 response (Fisher`s exact test. 1.00) and in the latter case survival for the 36 patients that received potentially curative treatment with respect to their age. Survival therefore is, not unexpectedly, linked with younger age but is not an explanation for the improved survival for the patients that respond towards TLR targeting. Overall, 37% of patients in the survival analysis were non-responders to TLR1/2 targeting as opposed to 39% in the whole population, implying that TLR1/2 response was evenly distributed also among the foremost elderly patients not eligible for intensive treatment.

       e) For the transcriptomics analysis presented in Figure 3, it is unclear which samples were used and what state they were used at. Are these TLR stimulated samples? If so, with which? If not, are these cultured? Viably frozen? Sorted for live blast cells? Or total pellets? Were all samples obtained, processed and treated in the same manner and same timeframe (i.e. sample storage between processing steps)?

We have added the requested information to the figure legend. A comment on storage time has also been added to the first chapter in the Materials and Methods section (lines 476-479).

            f) NPM1 mutations are referenced in several instances. How were these assessed? Is the data presented elsewhere?

This is a part of the routine testing of AML patients at our hospital; the data are available through the biobank.

            g) Line 265 references morphological analysis. How was this analyzed? Is the data presented elsewhere?

This was performed by histochemical staining and a part of the routine testing of AML patients at our hospital; the data are available through the biobank.

            h) Line 268 discusses clustering of 129 proteins for further analysis.  Where did this list come from?

We agree with the reviewer that additional information was needed. A short introduction to result section 2.5 was therefore added (lines 196-198).

            i) Paragraph starting line 282 discusses purity of the samples to >90% blasts. Is this data shown elsewhere? How was this determined?

This was determined by morphological examination and by flow cytometry; this information is also available through the biobank.

4. There are also several instances where comparison to previous literature (or description that this is an original finding) would put this data in perspective, for example in Figure 1 – how do these TLR responses compare to other studies of AML cells/AML cell lines/other primary samples?

In our present study we have a focus on demonstrating functional receptors that initiate downstream signaling, the expression of downstream pathway members and not only the receptor structure, the importance of TLRs with regard to communication through the local cytokine network (i.e. AML cell cytokine release) between AML cells and their AML-supporting non-leukemic cells in their common microenvironment, and associations between functional TLR signaling and protein levels. These aspects have not been addressed in previous studies. This is now briefly mentioned at the end of the Introduction and at the beginning of the discussion (lines 65/66 and lines 304/305, respectively). 

5. The result descriptions are difficult to follow and could use clarification/simplification (fewer compound sentences would help here).

a) One example is the paragraph starting on line 73 – the description of responders is needed and all relevant data is here, but it is hard to follow and could perhaps in a simpler format here and more of this information moved to the methods.

b) There are some grammatical errors that can be misleading and should be corrected.

c) and a few additional typographical errors (one example, line 180, referencing a previous chapter.

We have carefully controlled grammar and spelling throughout the manuscript, and we have had a focus on the Results section and tried to simplify the language.

6. The conclusions are ambiguous and overstretched in places, some points of confusion:

            a) The title emphasizes TLR1/2-NFkB signaling, however, the experimental data also includes stimulation for TLR4, 5 and 7/8 as well, which is appropriate for controls (I agree here with their inclusion), but which are given equal weight in the discussion and seem to have equal responses in the data presented (i.e. Figure 3). The discussion also focuses these other TLRs as well, so the emphasis on just TLR1/2 in the title seems limiting.

The title has been rewritten

            b) The conclusions stating that a number of factors are not correlated with TLR responses seem overstated given the patient cohort size (and lack of significant numbers with in specific cohorts).

An additional comment is now included at the end of chapter 2.3 (lines 146-148).

            c) The link of NFkB seems tenuous. More clarification of how the authors see NFkB regulation, or specific NFkB pathway members that are differentially regulated in this data set could be highlighted.

The five specific proteins that are upregulated in the TLR responders and that are associated with NFkB inhibition are now provided in the text (lines 204-206 and lines 533/534).

Reviewer 4 Report

What happens when normal cells (controls) are induced with TLR receptor inducers? Controls are lacking in these experiments. Is the upregulation of cytokines specific to AML blasts?  

Page 3, lines 74-76: Why was the response assessment done this way? Was the definition of responders and non-responders established before the experiments? 

Lines 110-112: Since numbers are limited, it would be helpful to provide ranges or 95% confidence intervals.

It would be useful to confirm results in an external cohort such as TCGA to confirm the prognostic impact of TLR receptor expression/activation.

Author Response

1. What happens when normal cells (controls) are induced with TLR receptor inducers? Controls are lacking in these experiments. Is the upregulation of cytokines specific to AML blasts? 

We have now added a new chapter in the Discussion section; a detailed discussion is presented based on new references; additional analyses of the available data are also presented. Our conclusion is that the present data can only be explained by an effect on the leukemic cells (lines 343-359).

2. Page 3, lines 74-76: Why was the response assessment done this way? Was the definition of responders and non-responders established before the experiments?

A more detailed description of our definitions has now been included in section 2.3 (lines 111-121). Please also see our response to reviewer 2 comment 2.2. We decided our strategy for analyses of our data when we designed the study. We want to keep a focus on cytokine/mediator profiles rather than single mediators, and we try to identify and characterize the main patient subsets and not additional minor subsets. Even though our study includes more patients than many other (maybe most?) experimental AML cell studies, we want to keep to this main strategy. We hope this can be acceptable both for reviewer 2 and reviewer 4.

3. Lines 110-112: Since numbers are limited, it would be helpful to provide ranges or 95% confidence intervals.

Confidence intervals were added (lines 166-171). However, 52% of the TLR1/2 responders were censored. Thus, the confidence interval could not be calculated for this particular case.

4. It would be useful to confirm results in an external cohort such as TCGA to confirm the prognostic impact of TLR receptor expression/activation.

We agree that it would definitely be useful to verify our results in an external cohort, such as the Cancer Genome Atlas Program. All parts of our study are based on detection of functional TLRs by cell culture and estimation of cytokine release responses. To the best of our knowledge such functional data are not available in this or any other database.

Our studies suggest that genomic/proteomic biomarkers should be considered to identify the patient subsets that differ in their responsiveness to TLR agonists; and preferably be done in a prospective way.

We agree that, if possible, it would have been an advantage to analyze an external cohort. This is also stated in our concluding comment. However, this comment is not stated as an absolute requirement. For this reason, we hope that our response to this single comment will not be decisive for the final editorial decision about our article.

Round  2

Reviewer 2 Report

The secretion of 20 different cytokines in response to stimulation with four toll-like receptor (TLR) ligands was monitored in AML samples and correlated with the occurrence of mutations in the nucleophosmin gene, patient survival and protein expression networks.

The revised manuscript and the authors’ replies to reviewer comments provided some important clarifications

1)            Although a high number of AML cell samples were available for stimulation by TLR agonists and analysis of a panel of cytokines, the description of TLR responses remains superficial and lacks reference values. Perhaps it would be worthwhile to also look at the cytokine concentrations achieved in the culture supernatants and compare them with reference cell lines. To which degree is the cytokine secretion a fixed property of the AML cells or influenced by clonal evolution or even laboratory procedures?

2)            It would be much more plausible to correlate the response to TLR stimulation with genetic or biological features than with patient survival. In this regard, I do not understand, why there is no Figure about the correlation of the response to TLR activation with the occurrence of mutations in the nucleophosmin gene.

3)            In the new version it became clear that only patients who received chemotherapy were included in the survival analysis and that the survival curves in Figs. S1 and 2 are meant to establish a correlation of the response to TLR stimulation with chemo-sensitivity. A much more direct way to show this relationship would be to test cellular survival of the AML samples to chemotherapeutic agents. Also correlation with the proliferation capacity described in ref. 24 could be of interest.

4)            I still have doubts that the response to TLR stimulation is linked to patient survival in a relevant manner. The numbers of patients in the responder and non-responder groups should be directly indicated in the survival curves. For the data shown in Fig. S1, the authors admit that the non-responder group is too small to allow the conclusions made. Why should this situation be improved, if the correlation is restricted to only one agonist, which does not elicit a response for most of the cytokines tested? Despite some clarifications, the authors’ conclusion stated in the title and abstract, namely that the survival of AML patients undergoing chemotherapy can be linked to the TLR response of their tumor cells, is not supported by sufficient evidence.

Reviewer 3 Report

The authors' responses to review comments are thorough and complete. I feel that all comments were adequately addressed and that the paper is significantly stronger as a result. Overall, I feel that this paper makes a significant contribution, completely describes the study methods, fairly states its conclusions, and is complete for publication as is.

One very minor comment remaining regarding methods.  For the non-AML-specific clinical audience, who are also interested in the broader implications of this manuscript, the questions regarding the methodology (original review comment 3f, g, and i) that the authors addressed as standard biobank testing, may still benefit from brief inclusion in the methods (something such as "standard biobank testing includes screening for NPM1 mutations, morphological analysis, and blast percentage"). Although, perhaps this is not necessary if the intended audience is exclusively clinical. 

Reviewer 4 Report

All my comments were adequately addressed.